# Does Supramolecular Gelation Require an External Trigger?

**DOI:** 10.3390/gels8120813

**Published:** 2022-12-10

**Authors:** Ruben Van Lommel, Julie Van Hooste, Johannes Vandaele, Gert Steurs, Tom Van der Donck, Frank De Proft, Susana Rocha, Dimitrios Sakellariou, Mercedes Alonso, Wim M. De Borggraeve

**Affiliations:** 1Molecular Design and Synthesis, Department of Chemistry, KU Leuven, Celestijnenlaan 200F, Box 2404, 3001 Leuven, Belgium; 2Eenheid Algemene Chemie (ALGC), Department of Chemistry, Vrije Universiteit Brussel (VUB), Pleinlaan 2, 1050 Brussels, Belgium; 3Molecular Imaging and Photonics, Department of Chemistry, KU Leuven, Celestijnenlaan 200F, Box 2404, 3001 Leuven, Belgium; 4Department of Materials Engineering, KU Leuven, Kasteelpark Arenberg 44, 3001 Leuven, Belgium; 5Center for Membrane Separations, Adsorption, Catalysis and Spectroscopy for Sustainable Solutions (cMACS), Department of Microbial and Molecular Systems (M2S), KU Leuven, Celestijnenlaan 200F, Box 2454, 3001 Leuven, Belgium

**Keywords:** gels, self-assembly, single particle tracking, external stimuli, LMWG

## Abstract

The supramolecular gelation of small molecules is typically preceded by an external stimulus to trigger the self-assembly. The need for this trigger stems from the metastable nature of most supramolecular gels and can limit their applicability. Herein, we present a small urea-based molecule that spontaneously forms a stable hydrogel by simple mixing without the addition of an external trigger. Single particle tracking experiments and observations made from scanning electron microscopy indicated that triggerless gelation occurred in a similar fashion as the archetypical heat-triggered gelation. These results could stimulate the search for other supramolecular hydrogels that can be obtained by simple mixing. Furthermore, the mechanism of the heat-triggered supramolecular gelation was elucidated by a combination of molecular dynamics simulations and quantitative NMR experiments. Surprisingly, hydrogelation seemingly occurs via a stepwise self-assembly in which spherical nanoparticles mature into an entangled fibrillary network.

## 1. Introduction

Low molecular weight molecule-based supramolecular gels (LMWGs) have drawn considerable attention in both academia and industry over the past decades as usable soft materials [1,2,3,4]. To achieve the gelation of a solvent, small gelator molecules are added to the medium in amounts surpassing the critical gelation concentration, usually resulting in a saturated suspension. Typically, an external stimulus is applied to solubilize the gelator. Finally, by means of a trigger, the solution is reverted to a supersaturated state, causing the non-equilibrium self-assembly of the gelator molecules into an anisotropic nano-architecture. Although several 0D, 1D, 2D, and 3D structures have been reported, the most common nano-architectures observed are the 1-dimensional nanofibers [5]. Through the entanglement of these architectures, a self-assembled network is formed that spans and immobilizes the medium, causing the characteristic gel features. This gelation process is affected by a variety of external parameters such as the concentration of the gelator and the type of trigger used. The trigger can be understood as a change of external or internal conditions that induces the sol-to-gel transition (i.e., the self-assembly of the solubilized gelator toward the supramolecular gel). Although at first sight this process might seem trivial, its complexity is evidenced by the myriad of triggers that are reported in the literature including a heating-and-cooling cycle [6], pH [7], light [8], sonication [9,10,11], solvent-switching [12], chemical reactions [13] and ion- and metal-induced gelation [14,15]. Moreover, recent studies indicate supramolecular gelation to be a process controlled by kinetics or thermodynamics, depending on the system. Indeed, while the supramolecular gel state was generally believed to be a kinetically-trapped metastable state because of the spontaneous gel-to-crystal transitions observed for multiple systems, a recent theoretical investigation led by Tuttle and Ulijn revealed that supramolecular gel fibers could also be at thermodynamic equilibrium [16,17]. Clearly, further fundamental studies are required to reach a general overview of the kinetic and thermodynamic effects that drive supramolecular gelation.

Because these materials owe their structure to weak noncovalent interactions, they are often sensitive toward external stimuli. As such, applications envisioned for LMWGs frequently exploit this responsiveness as smart materials [15,18,19,20,21]. Nevertheless, in some cases, the sensitivity of LMWGs constitutes a major drawback, making them somewhat less ‘smart’. LMWGs designed for biomedical applications are required to be chemically inert toward the electrolytes present in the body and should have an adequate stability across the physiological temperature and pH range [22,23]. Moreover, in contrast to their polymeric counterparts, most supramolecular LMWGs are characterized by weak mechanical properties, which limit their usability in real life applications. Therefore, state-of-the-art research is focused on fine-tuning the mechanical response of these materials and gaining spatiotemporal control over the sol-to-gel transition [24,25,26,27,28].

To achieve this, the field has turned its attention toward the gelation process itself [29,30]. This either means tailoring how the sol-to-gel trigger is applied, or changing the nature of the trigger altogether. In this regard, LMWGs that are sensitive to multiple external stimuli have gained substantial interest [31,32,33,34]. These multi-stimuli responsive LMWGs can have their sol-to-gel transition induced by applying various external triggers. Depending on the type of gelation trigger used, supramolecular gels with different material properties can be obtained, notwithstanding that the chemical composition of the material remains the same. While an extensive library of external triggers has already been reported to achieve the sol-to-gel transition, to the best of our knowledge, the conceivably most elementary manner to achieve gelation (i.e., by the simple addition of the pure gelator to the medium at room temperature), without altering external or internal conditions afterward, has not been reported thus far [1]. The requirement for such a trigger to accomplish supramolecular gelation can most likely be explained by the limited solubility of the gelator in the medium, combined with the metastable nature of most supramolecular gels [35,36]. With this in mind, we became particularly interested in the bis-urea-based hydrogelator depicted in Figure 1 (**p2*o*1**). Historically, urea derivatives form an important subclass of supramolecular gels and this molecule served as one of the model compounds for the development of our multiscale computational approach to study the noncovalent interactions steering supramolecular gelation [37,38,39,40,41,42]. Additionally, we showed that **p2*o*1** had excellent hydrogelation abilities, being able to fully immobilize water at a concentration as low as 0.4% *w*/*v* by means of a heating-and-cooling cycle (i.e., heat-triggering) [43]. More compelling however, is that exploratory observations suggested that this gelator is able to immobilize water by simple addition to the medium without changing any internal or external conditions afterward (i.e., triggerless) [44]. In this work, we set out to elucidate the ability of **p2*o*1** to gel water without the need for an external trigger as well as to understand the underlying mechanistic features that enable this.

## 2. Results and Discussion

Visual inspection showed that a self-supporting hydrogel could be obtained in a matter of days by simply combining **p2*o*1** with water at a concentration of 1.0% *w*/*v* without using an external trigger (triggerless gelation). Moreover, as time progressed, the material retained its stability and became significantly more opaque (Figure 1a). To gain a deeper understanding of the changes that spontaneously occurred at the microscopic level, suspensions of **p2*o*1** in water (1.0% *w*/*v*) were quenched and freeze-dried after 1 h and after 14 days, respectively. Back-scattered electron (BSE) images were taken from the corresponding xerogels (Figure 1b and Appendix A). While removing water by means of freeze-drying reduces potential drying artefacts in the material, we will refrain ourselves from making quantitative conclusions based on these images [45]. Upon closer inspection, two different architectures could be observed: nanospheres and fibers. After 1 h, a significant amount of nanospheres was detected. After 14 days, however, a network formed that spanned a larger volume and was composed mostly out of nanofibers. In some parts of the material, it could be observed that the fibers were connected to clusters of nanospheres, similar to a grapevine structure. From these results, we hypothesized that over time, a transition occurs from nanospheres toward a self-assembled fibrillar network when **p2*o*1** is combined with water.

For comparison purposes, the corresponding heat-triggered hydrogelation of **p2*o*1** was also investigated. During heat-triggered gelation, the gelator is first solubilized in water by increasing the temperature, after which it is allowed to cool to room temperature. Simple inspection of the sample during cooling revealed a remarkable gelation mechanism (Figure 1c and Appendix A). Indeed, when a sample of **p2*o*1** in water (1.0% *w*/*v*) was heated to 100 °C in a closed vial, a clear solution was obtained, as the solubility of the gelator increased. When removing this sample from the heating element to the bench, the material immediately turned opaque (after 1 min). This usually indicates gelation to have occurred; however, at this stage, the material is not self-supporting and is characterized by liquid features. Surprisingly, after 15 min, the sample returned to a transparent state and became substantially more viscous. In a later stage, the material turned opaque once more and a self-supporting hydrogel was obtained. The full gelation mechanism took place in a timespan of approximately 25 min. BSE images of the corresponding xerogel revealed a network of sheets, composed of a high density of entangled fibers (Figure 1d). In some areas, nanospheres were present, similar to the structures observed during the spontaneous gelation. Interestingly, a gel network with comparable features could be obtained when a change in pH or sonication was used as the trigger (Appendix A).

Single particle tracking experiments using fluorescence microscopy allowed us to monitor the microscale changes that occurred during the triggerless and heat-triggered gelation. After doping the sample with chemically inert fluorescent beads, it was possible to follow the movement of these particles with nanometer precision in 3-dimensions using an in-house-built multifocal plane microscope [46]. This allowed us to determine the trajectories of each individual particle over time. From there, based on the mean-square displacement of the beads, their diffusion coefficient (D) was obtained, from which the viscosity of the medium was also derived using the Stokes–Einstein relationship [46]. A more detailed explanation of the experimental setup and results can be found in the methods section and in the Appendix A. In general, the evolution of the diffusion coefficient of the beads and of the viscosity of the medium agreed with a transition toward a gel phase for the heat-triggered and triggerless gelation (Figure 1e and Appendix A). Immediately after removing the heating element (at t = 0), the diffusion coefficient of the beads started to decrease, which can be assumed as a direct consequence of the associated decrease in the temperature of the sample during the experiment. However, only after 15 min, a significant increase in the medium viscosity was observed, while after 25 min, the beads were fully immobilized by the gel network, hinting at complete gelation. Note that the medium’s viscosity increased by several hundred cP during the heat-triggered gelation, meaning that the inherent temperature dependence of the viscosity of pure water could not be the main reason for the observed trends [47]. This immobilization process could also be observed in real-time from the dynamic fluorescent spectroscopic measurements (Appendix A). Moreover, during the triggerless gelation, the fluorescent beads seemingly clustered together. This clustering behavior might be caused by the separated pockets that are formed during gelation, which is a key feature of a porous gel network (Figure 1f).

While the heat-triggered and triggerless gelation process clearly showed similar trends in the viscosity and morphological changes of the material, it is important to note that the time scale at which gelation took place was considerably different. During a heat-triggered gelation, the full process took approximately 25 min. During triggerless gelation, early local viscosity changes occurred only after 6 h. On the other hand, immobilization of larger volumes relying on triggerless gelation could take up to several days, depending on the volume of the sample and concentration of the gelator (Appendix A).

All-atom molecular dynamic (MD) simulations allowed us to investigate the early aggregation phase of **p2*o*1** in water. Recently, we developed four MD-derived descriptors that can help analyze the evolution of the resulting MD trajectories: (i) rSASA, which values aggregation, (ii) HB%, which quantifies the hydrogen bonding propensity between the gelator molecules, (iii) the relative end-to-end distance rH, which measures the flexibility and conformational preference of the individual gelators; and (iv) the shape factor F, which describes the shape of the aggregates [48]. In total, three separate MD simulations with a total simulation time of 150 nanoseconds were initiated, starting from a fully dispersed state by setting the minimum starting distance between the gelators at 3.0 Å (Appendix A).

During the simulation, the aggregation of **p2*o*1** in water occurred instantly, as evidenced by the rapid increase in the number of hydrogen bonds between the gelator molecules and the decrease in rSASA value in the first 20 nanoseconds of each simulation (Figure 2). Following this evolution, the aggregates clustered together, forming a single larger nano-architecture. This translated into a further decrease in rSASA, reaching a value of approximately 0.3. Furthermore, as the shape factor F was computed based on the radius of gyration, which itself is dependent on the moment of inertia of the molecular structure, a sharp decrease of F was observed when all of the larger aggregates had been incorporated into a single nano-architecture. Notably at this stage, single **p2*o*1** molecules dynamically attached and detached from the nano-architecture. During the entire process, the relative end-to-end distance rH fluctuated around a value of 0.7. This means that the gelator molecules prefer an extended conformation. The MD simulations and the associated evolution of the descriptors allowed us to shed light on the early aggregation phase. First, the solubilized gelators rapidly aggregated in water, initiated by the tendency of **p2*o*1** to form hydrogen bonds amongst one another. Second, aggregates of the gelator tended to combine and shape into a larger nano-architecture, minimizing the contact area with the solvent and thus the hydrophobic interactions.

To elucidate the heat-triggered hydrogelation mechanism of **p2*o*1**, variable temperature (VT) ^1^H-NMR experiments were carried out. By studying the change in chemical shifts, the NMR relaxation times, or integrated intensities of the signals, valuable information can be extracted concerning the self-assembly behavior of the gelator. Indeed, as a sol-to-gel transition occurs, it can be expected that the internal mobility of the gelator molecules significantly decreases. This often translates into a decreased transverse relaxation time (*T_2_*) and increased peak-broadening or reduction in the integration intensities of the resonance signals [49,50,51]. Moreover, as the self-assembly of the gelator molecules relies on intermolecular non-covalent interactions, a change in chemical shifts can often be observed, caused by the alteration in the chemical environment during the sol-to-gel transition [52]. Nevertheless, due to the viscoelastic nature of the soft material, it is not always straightforward to conclude in which phase the gelator molecules that are generating the NMR signals find themselves [53]. Consequently, the results from these experiments must be carefully scrutinized.

A VT-^1^H-NMR experiment of the **p2*o*1** gel at a concentration of 1.0% *w*/*v* in H_2_O:D_2_O (9:1) was performed, starting from a gelled sample, which was heated to 85 °C to induce a gel-to-sol transition (Figure 3a, Appendix A). At 25 °C, well-resolved ^1^H-NMR peaks were visible. This is remarkable considering the fact that at this temperature the sample is fully gelled, which usually results in signal broadening [54]. Because it was also possible to detect ^13^C signals and the urea nitrogen atoms through ^1^H-^15^N HSQC experiments, a complete peak assignment of the spectra was possible (Appendix A). When the temperature increased, the gel-to-sol transition occurred. A notable observation was that upon heating, a set of broader peaks started appearing from approximately 55 °C onward. Importantly, when the experiment was reversed (i.e., a VT-NMR experiment where a fully solubilized sample is cooled to room temperature), the same peaks emerged in the temperature range from 70 °C to 40 °C (Appendix A). We concluded that both sets of signals stemmed from **p2*o*1**, but in a different material phase, with the sharper, well-resolved peaks originating from molecules in solution, and the broader set of signals originating from mobile aggregated structures. This conclusion is based on the observations that:(i)For each narrow signal, a corresponding broader signal appeared upfield with a similar ^1^H-^13^C HSQC cross-peak pattern (Appendix A). Note that for the urea hydrogen atoms H-11 and H-9, the original narrow and broader signal (partially) overlapped with one another.(ii)The ^1^H-^1^H NOESY spectrum of the gel at 60 °C showed several cross-peaks for the well-resolved signals, which can all be accounted for by intramolecular proximity. On the other hand, the broader set of signals showed cross-peaks between the aromatic (7.0–6.7 ppm) and aliphatic (3.95, 3.24 and 2.62 ppm) region, which points toward intermolecular interactions and/or conformational changes (Appendix A).(iii)Upon decreasing the concentration of **p2*o*1** in the sample, the broader signals were absent, highlighting the importance of intermolecular interactions (Appendix A).

Some observations indicate the well-resolved peaks associated with the solvated **p2*o*1** molecules also contained information about a larger aggregated structure. First, the sharp resonance signals for a gelled sample were characterized by an increased rotational correlation time compared to the corresponding signals of a sample below the critical gel concentration (Appendix A). Additionally, cross-peaks in the NOESY spectrum of the gel sample were characterized by a sign, similar to the diagonal peaks. This, together with a relatively low mixing time of 0.1 s, suggests that the signals arose from a slow-motion regime (Appendix A). An explanation for these seemingly contradicting results can be put forward by assuming a thermodynamic equilibrium between the gelators in solution and the gelators in the aggregated phase. This equilibrium might be slow with respect to the chemical shift difference, explaining why the gelators in solution appeared as highly resolved peaks, but fast with respect to the observed relaxation times, resulting in NOESY cross-peaks characterized by the same sign as the diagonal peaks and increased rotational correlation times [53].

Having established the nature of all peaks, Bruker’s external quantification tool ERETIC2 was used to quantify the gelation mechanism [55]. qNMR through ERETIC2 is straightforward and omits the need for an internal standard (which could affect the gelation mechanism) by correlating the absolute intensities of two spectra from different samples. This tool is based on the previously published PULCON method [56]. The experiment showed that between 20 °C and 80 °C, some of the gelator molecules were solution-state NMR silent (Figure 3c). This implies that besides the gelator in solution and the mobile aggregates, a third molecular phase of **p2*o*1** should be considered, which most likely corresponds to the immobilized and solution-state NMR silent gel phase.

Based on these VT-qNMR experiments and recalling the observations and experiments that were previously described in this work, we believe that the heat-triggered hydrogelation mechanism of **p2*o*1** can be described as follows: Gelator molecules at high temperature become dispersed and solubilized. When the temperature decreases, **p2*o*1** will start to assemble into small mobile nanospheres. At this stage, the bulk material is still characterized by liquid properties. In a second step, these spherical aggregates cluster together. Subsequently, these clusters reshape into nanofibers that entangle and form the immobilized gel network. This is accompanied by a drastic increase in viscosity. At room temperature (20 °C) the majority of gelator molecules is immobilized and incorporated into the network, although a small amount of **p2*o*1** remains in solution, providing a solubility of 2.0 ± 0.5 mM in water (Figure 3c and Appendix A). Usually, it is assumed that single gelator molecules are gradually incorporated into the immobilized network through an isodesmic or cooperative chain growth mechanism [57]. Only recently has the group of Gazit introduced a stepwise assembly process as described above, for a minimalistic peptide-based system [58,59]. As such, we wonder whether other supramolecular gelators undergo a similar gelation mechanism.

## 3. Conclusions

In this work, we set out to better understand the heat-triggered gelation mechanism of the urea-based supramolecular hydrogel of **p2*o*1**. Through a combination of quantitative NMR experiments, molecular dynamics simulations and electron microscopy imaging, an uncommon stepwise self-assembly mechanism was revealed in which spherical particles gradually mature into a nanofibrous network. Furthermore, single particle tracking experiments provided evidence that a self-supporting hydrogel could be formed without the need for any external trigger by adding the pure gelator molecule to the medium. While triggerless gelation took place over a significantly longer timespan compared to its heat-triggered equivalent, it also implies an increased robustness of the gel phase, which could contribute to the usefulness of these materials. We hope that this work incentivizes the field to develop other supramolecular gels that can be formed without an external trigger and are characterized by improved kinetics.

## 4. Materials and Methods

### 4.1. Synthesis, Characterization and Hydrogelation of **p2o1**

The synthesis of the bis-urea based gelator has previously been described in the literature and can be performed on gram scale in yields up to 66% [43]. The identity of the compound was confirmed via ^1^H and ^13^C NMR as well as ESI-MS and FT-IR. Hydrogelation of **p2*o*1** was carried out according to the specified gelation process and always in water of Milli-Q grade.

ESI-MS (*m/z*): [M + H^+^] calculated for C_24_H_29_N_6_O_2_^+^, 433.2 g/mol; found, 433.2 g/mol.

^1^H NMR (300 MHz, DMSO-*d*_6_, p_h._): 8.48 (2H, ddd, *J* = 5.1, 1.9, 0.9 Hz), 7.70 (2H, d, *J* = 7.7, 1.9 Hz), 7.29–7.14 (5H, m), 7.11–7.01 (3H, m), 6.37 (2H, t, *J* = 5.7 Hz), 5.94 (2H, t, *J* = 5.8 Hz), 4.18 (4H, d, *J* = 5.8 Hz), 3.39 (4H, td, *J* = 6.9, 5.8 Hz), 2.85 (4H, t, *J* = 6.8 Hz).

^13^C NMR (100 MHz, DMSO-*d*_6_, δ_C_): 159.46, 157.95, 149.00, 140.82, 136.37, 128.11, 125.74, 125.30, 123.18, 121.37, 42.91, 39.14, 38.25.

FTIR (cm^−1^): 3317 (N-H stretch), 1619 (C=O stretch).

### 4.2. NMR Experiments on Gelled Sample of **p2o1**

Gelled samples for the NMR experiments were prepared by heating a screw capped vial containing the gelator **p2*o*1** in a H_2_O:D_2_O mixture (varying ratios depending on the experiment can be retrieved from Appendix A) to 120 °C using a copper heating block. In general, the concentration of the gelator in H_2_O:D_2_O was set to 1.0% *w*/*v*. The presence of D_2_O was required for field-frequency locking. After 5 min of heating, when the gelator was fully dissolved, the warm solution was quickly added into a 5 mm NMR tube through a syringe. The gel was obtained after 30 min of cooling to room temperature. H_2_O of ultrapure Milli-Q grade was used in all experiments. Furthermore, all NMR spectra were recorded on a Bruker Avance II+ 600 spectrometer operating at 14.1 Tesla with a 5 mm TXI H-C/N-D probe (for all ^1^H-detected experiments) or a 5 mm BBO (^31^P-^103^Ag)-^1^H/D probe (for ^13^C-detected experiments). A more detailed description of all NMR experiments that were performed in this study can be found in the Appendix A.

### 4.3. Molecular Dynamics Simulations

Simulations were performed with the open-source GROMACS software using the CHARMM27 force field, and the TIP3P water model and parameters for the gelator came from the Swissparam service [60,61,62,63]. A more detailed description can be found in the Appendix A.

### 4.4. Scanning Electron Microscopy Imaging

Xerogels were obtained by freeze-drying the corresponding hydrogel. Samples of this xerogel were coated with Pt (5 nm) using a Quorum Q150T S coater system, while the BSE and SEM images were taken using a FEI Nova NanoSEM 450 FEG SEM operating at a landing energy voltage of 5 kV (using Beam Deceleration, 4 kV). More details can be found in the Appendix A.

### 4.5. Single Particle Tracking Experiments

Samples were freshly prepared before each measurement. A precise amount of **p2*o*1** in powder form was weighed on a balance directly in a LoBind tube (Eppendorf) and mixed with 1 mL of Milli-Q water to obtain the desired final concentration. The solution was then vortexed for 10 s and transferred to a covered round glass bottom dish (ThermoFisher). For 1 mL of Milli-Q water, 30 µL of fluorescent beads was added (200 nm radius, FluoSphere, 505/515, carboxylate modified, Thermo Fisher Scientific (Waltham, MA, USA) with a final dilution of 1:3000. For spontaneous gel forming experiments, the sample was directly mounted on the microscope in a completely dark room. For heat-induced experiments, the sample was first heated up to 100 °C on a heating plate for 5 min before mounting on the microscope. All samples were prepared fresh and in triplicate.

The microscope used in this research was a custom built setup that has been fully described in previous work by Louis et al. [46]. Briefly, fluorescent beads were excited using a 488 nm laser (SpectraPhysics) via a water-immersion objective (UPlanSApo60XW, Olympus (Tokyo, Japan)). Emission light passed through an emission filter (HQ535/50M, Chroma (Rockingham, VT, USA)), a special set of lenses, and a proprietary prism acting as a beam splitter, allowing for multi-plane imaging. The final signal was captured using two sCMOS cameras (Orca Flash 4.0, Hamamatsu Photonics Inc. (Hamamatsu, Japan)). The hardware synchronization, allowing for fast imaging acquisition, was conducted by using a National Instruments Board (NI, USB-6343). The operating software was completely custom made in Micro-Manager [64]. All data were processed using an in-house routine built in MATLAB, which can be freely accessed on Github (https://github.com/CamachoDejay/polymer3D (accessed on 7 December 2022)).

## Data Availability

Data are contained within the article or Appendix A.

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
