# Peer review of "Does Supramolecular Gelation Require an External Trigger?"

_gels, 2022, doi:10.3390/gels8120813_

Round 1
Reviewer 1 Report
In this paper, the authors show their efforts to make supramolecular hydrogels without using any triggers. The hydrogels are formed by dispersing a small urea-based molecular gelator in water immediately that does not require any external stimuli. The gelation mechanism was explained by structure analysis on the hydrogel networks using SEM, molecular dynamics simulation, and quantitative NMR measurements. In my view of point, this is paper is lack of novelty, gelation with using triggers is actually available for many gelators, the only problem is the gelation process is rather slow because of the slow transfer of the gelator from bulk to the solvent and thereby self-assembly into the supramolecular structures. The authors declare that they have revealed the mechanism of the gelation using MD and NMR, but to me, it is not clear what the mechanism exactly is, i.e. why the urea-based molecular gelator can self-assembly without using triggers remains unclear to the readers. Furthermore, the data organization is confusing. I do not understand why the author compare the spontaneous gelation process with that triggered by heat. They should focus on the gelation without triggers; or at least the data related to use of triggers should be moved to ESI. Therefore, I cannot recommend for publication of this paper in Gels.
By the way, some minor issues should be improved: the supporting movie S2 is wrongly cited in the paper I think; the scale bars in fig.1 are hard to read; as shown in Fig.1e, why the particle diffusion efficiency for the sample of heat-triggered gelation is lower than that of spontaneous gelation, because heat will of course increase the diffusion of objects.
Author Response
Note that the referee comments are in green and our answers in black.
In this paper, the authors show their efforts to make supramolecular hydrogels without using any triggers. The hydrogels are formed by dispersing a small urea-based molecular gelator in water immediately that does not require any external stimuli. The gelation mechanism was explained by structure analysis on the hydrogel networks using SEM, molecular dynamics simulation, and quantitative NMR measurements. In my view of point, this is paper is lack of novelty, gelation with using triggers is actually available for many gelators, the only problem is the gelation process is rather slow because of the slow transfer of the gelator from bulk to the solvent and thereby self-assembly into the supramolecular structures.
We thank the referee for providing us with honest remarks and expressing a concern about the novelty of the work. We are aware that the process of supramolecular gelation using an external trigger is a common, well studied phenomenon. Triggerless supramolecular gelation however, we have to the best of our knowledge, not seen reported thus far. The system that comes closest to triggerless gelation was reported in 2019 (Mater. Chem. Front., 2019, 3, 2637-2646), where gelation is triggered by DMSO and DMF capturing atmospheric water. The lack of reporting triggerless gelation was also explicitly mentioned in an impactfull review in 2017 (Chem, 2017, 3, 390–410). We quote from the paper: “To the best of our knowledge, there are no examples where a molecule is simply added to a solvent and gelation results…” We do want to emphasize that we are discussing supramolecular gelation and not the gelation of a solvent by a polymeric system, for which triggerless gelation might be more commonly described. Besides SEM, MD simulations and NMR spectroscopy, we have also used advanced single particle tracking experiments to track the morphological changes that occur during supramolecular gelation. Tracking this supramolecular process in-situ without having a significant impact on the system during the experiment has been a challenge in the field for many years. The introduction of single particle tracking experiments in the field of supramolecular gels thus also increases the novelty of the work.
The authors declare that they have revealed the mechanism of the gelation using MD and NMR, but to me, it is not clear what the mechanism exactly is, i.e. why the urea-based molecular gelator can self-assembly without using triggers remains unclear to the readers.
In essence, the manuscript is a curiosity-driven work that provides the field with two uncommon observations for supramolecular gels, further developing our fundamental understanding of these systems. First, we prove that supramolecular gelation can occur without the need for an external trigger. This has major implications on the general understanding that the supramolecular gel state should always be a kinetically trapped metastable state. The second uncommon observation is the mechanism of the heat-triggered gelation that occurs in a stepwise fashion. With our experiments, we cannot conclusively state that the triggerless gelation proceeds through the same mechanism as the heat-triggered gelation. Furthermore, we cannot give a definite reason why this specific urea-based molecule can self-assemble into a supramolecular gel without a trigger, only that it does. We acknowledge that some parts of the text were not well structured or did not fully reveal that the triggerless gelation and elucidation of the mechanism are two separate parts of the manuscript. Hence, this has been made clearer by slighlty adapting the abstract and introduction and by emphasizing the difference between the heat-triggered and triggerless system throughout the manuscript.
Furthermore, the data organization is confusing. I do not understand why the author compare the spontaneous gelation process with that triggered by heat. They should focus on the gelation without triggers; or at least the data related to use of triggers should be moved to ESI. Therefore, I cannot recommend for publication of this paper in Gels.
We thank the referee for this comment, but do consider the experiments on the heat-triggered system crucial for the manuscript. In the first part of the manuscript, a comparative study between the triggerless and archetypical heat-triggered supramolecular gel is performed based on SEM imaging and single-particle tracking experiments. As thermal-triggered supramolecular gels have been extensively investigated in the past decade, this allows us to validate that the triggerless process results in a material with similar characteristics as the thermal-triggered material, e.g. a nanofibrous network that immobilizes the content of the material. Or in other words, these experiments are necessary to establish that a supramolecular gel can be formed by simple addition of the gelator to the medium, rather than only relying on the qualitative vial-inversion test. Moreover, as indicated in the previous response, a second part of the manuscript discusses the uncommon mechanism of heat-triggered gelation. While we agree with the referee that it would be very valuable to also investigate the mechanism of the triggerless supramolecular gelation, its slow kinetics, heterogeneous nature and sensitivity towards external vibrations made the NMR experiments impossible, which are crucial to investigate the mechanism in-situ.
By the way, some minor issues should be improved: the supporting movie S2 is wrongly cited in the paper I think; the scale bars in Fig.1 are hard to read; as shown in Fig.1e, why the particle diffusion efficiency for the sample of heat-triggered gelation is lower than that of spontaneous gelation, because heat will of course increase the diffusion of objects.
We checked the issues mentioned by the referee and took the following actions:
1) We re-checked the reference to supporting movie S2, which visualizes the immobilization of the fluorescent beads. In the main manuscript, the references to supporting movie S2 seem correct. Hence, if there were no issues with the upload of the movie, this issue should be resolved.
2) We agree that the scale-bar in Fig.1 are hard to read. To solve this issue we have increased the width of the scale bar, the size of the text, changed the colour from red to white and increased the overall size of the figure in the manuscript.
3) Indeed heat is expected to increase the particle diffusion. But what should be considered in the experiment that generated the graph in Fig. 1e is the following: 1) The measurement for heat-triggered gelation starts when the external heating is turned off. Because of the size and dimensions of the samle, cooling to room temperature is achieved rather quickly. 2) For the heat-triggered system, initial self-assembly occurs almost instantly (see MV_S1, sample becomes opaque right after heating is stopped). This self-assembly might already restrict the diffusion of the particles, explaining why the diffusion for heat-triggered gelation is lower than that of spontaneous gelation. The latter taking several hours before gelation takes place.
Reviewer 2 Report
This is a very interesting work. This manuscript describes gelation mechanism of the urea based supramolecular hydrogelator p201, mainly to reveal an unusual mechanism of gradual self-assembly of spherical particles maturing into nanofibrous networks.
However, there are still some concerns here.
(1) The authors propose that this gel is able to immobilize water by simply adding it to the medium without the addition of an external trigger. However, I am curious as to why the authors have spent so much time studying the mechanism of thermal triggering of the gel?
(2) In Page 2, Line 67-69, the authors describe a method to achieve gelation by adding pure colloids to the medium at room temperature, which has not been reported so far. However, there have been reports describing Supramolecular Hydrogelation of Urea-Based Gelators. (Ref.1: Chempluschem. 2020, 85(2):267-276.; Ref. 2: Chemical Communications, 2019, 55(51):7323-7326.; Ref.3: Chemistry - An Asian Journal, 2018, 13(8):929-933.)
(3) I am curious about the shear rheology of the hydrogel and the difference in the gelling ability of the supramolecular hydrogel for different concentrations of urea derivative.
Author Response
This is a very interesting work. This manuscript describes gelation mechanism of the urea based supramolecular hydrogelator p201, mainly to reveal an unusual mechanism of gradual self-assembly of spherical particles maturing into nanofibrous networks.
However, there are still some concerns here.
- The authors propose that this gel is able to immobilize water by simply adding it to the medium without the addition of an external trigger. However, I am curious as to why the authors have spent so much time studying the mechanism of thermal triggering of the gel?
We thank the referee for this valid comment that requires further clarification. In the first part of the manuscript, a comparative study between the triggerless and archetypical thermal-triggered supramolecular gel is performed based on SEM imaging and single-particle tracking experiments. As thermal-triggered supramolecular gels have been extensively investigated in the past decade, this allows us to validate that the triggerless process results in a material with similar characteristics as the thermal-triggered material, e.g. a nanofibrous network that immobilizes the content of the material. Or in other words, these experiments are necessary to establish that a supramolecular gel can be formed by simple addition of the gelator to the medium, rather than only relying on the qualitative vial-inversion test. In the second part of the manuscript, we dive into the mechanism of thermal-triggered (heat-triggered) supramolecular gelation. While we agree with the referee that it would be highly interesting to probe the mechanism of triggerless gelation, unfortunately, because of the slow kinetics of the process in combination with its heterogeneous character and sensitivity towards external vibrations, we were unable to collect similar NMR experiments to probe the triggerless gelation mechanism. Hence, we were only able to perform these experiments and probe the mechanism using the faster thermal-triggered gelation, which revealed an intriguing process that we felt was worth mentioning in the manuscript.
- In Page 2, Line 67-69, the authors describe a method to achieve gelation by adding pure colloids to the medium at room temperature, which has not been reported so far. However, there have been reports describing Supramolecular Hydrogelation of Urea-Based Gelators. (Ref.1: Chempluschem. 2020, 85(2):267-276.; Ref. 2: Chemical Communications, 2019, 55(51):7323-7326.; Ref.3: Chemistry - An Asian Journal, 2018, 13(8):929-933.)
In retrospect, we agree with the referee that the introduction could and should mention the historical and general importance of urea-based low molecular weight gelators. Accordingly, we added a short sentence to the introduction and added some key references (including the ones proposed by the referee) to guide the interested reader to useful literature on the use of urea derivatives as low molecular weight (hydro)-gelators.
- I am curious about the shear rheology of the hydrogel and the difference in the gelling ability of the supramolecular hydrogel for different concentrations of urea derivative.
Determining the mechanical properties of the supramolecular gel and test the influence of both the concentration as well as the type of trigger (thermal versus triggerless) through dynamic rheological analyses would indeed be interesting. In the past, our group has performed such measurements to characterize the supramolecular gels that were developed in the lab. (ChemPlusChem, 2020, 85(2):267-276 and Chemical Communications, 2019, 55(51):7323-7326). Herein, we did not perform any rheology measurements for two reasons: 1) We no longer have access to a dynamic shear rheometer and 2) the ‘triggerless’ supramolecular hydrogel is extremely fragile, making the setup of such an experiment arduous. This is also one of the reasons why we opted to perform single-particle tracking experiments as a ‘micro-rheology’ approach to still gain insights into the mechanical aspects of our materials i.e. the viscosity changes.
Reviewer 3 Report
The manuscript reports on triggerless gel system composed of water and a small molecule gelator.
The introduction lacks clear definition of what is considered to be a trigger and what is considered to be triggerless. Additionally, as gelation involves an intricate melange of thermodynamic as well as kinetic effects, I would expect the introduction to clearly outline the important parameters.
Heat trigger is also not defined. Figure 1 describes a system that gels much faster at elevated temperature than at room temperature. Is this a surprise? Can this be explained using normal kinetic and thermodynamic terms?Why do the authors claim there is no heat at 22C?!
It is my impression that the work is worth publishing but the manuscript is still half baked.
Author Response
The manuscript reports on triggerless gel system composed of water and a small molecule gelator.
The introduction lacks clear definition of what is considered to be a trigger and what is considered to be triggerless.
We appreciate the referee pointing out the lack of clarity regarding the definition of a trigger in our current manuscript. We, and others in the field of supramolecular gels, consider a trigger to be any change of internal or external conditions that induce a transition from a solution to a gel state. Triggerless can thus be understood as the absence of any change of conditions. To make this clearer in the manuscript and state the difference between the triggerless and heat-triggered gelation process, we have added/adapted the following paragraphs in the introduction:
“ … This gelation process is affected by a variety of external parameters, such as the concentration of the gelator and the type of trigger used. The trigger can be understood as a change of external or internal conditions that induces the sol-to-gel transition, i.e. the self-assembly of the solubilized gelator towards the supramolecular gel. Although at first sight this process might seem trivial, its complexity is evidenced by the myriad of triggers that are reported in literature, including: a heating-and-cooling cycle [6], pH [7], light [8], sonication [9-11], solvent-switching [12], chemical reactions [13] and ion- and metal-induced gelation [14,15]. “
and
“ … Additionally, we showed that p2o1 had excellent hydrogelation abilities, being able to fully immobilize water at a concentration as low as 0.4% w/V by means of a heating-and-cooling cycle (i.e. heat-triggering) [43]. More compelling however, is that exploratory observations suggested that this gelator is able to immobilize water by simple addition to the medium without changing any internal or external conditions afterwards (i.e. triggerless). “
Additionally, as gelation involves an intricate melange of thermodynamic as well as kinetic effects, I would expect the introduction to clearly outline the important parameters.
The thermodynamic and kinetic effects during supramolecular gelation remain a topical debate, not having reached a common consensus yet. Indeed, historically the supramolecular gel state was considered a kinetically driven metastable state. This train of thought was derived from the increase in studies reporting a spontaneous transition from gel-to-crystal as well as the observation that the properties of the gel could be dependent on the gelation process. Or in other words, different gel states were path-dependent. However, a recent theoretical study published by Tuttle and Ulijn (ACS Nano 2016, 10, 2, 2661–2668) revealed that supramolecular gel fibres could be thermodynamically stable. While triggerless gelation is indicative of a thermodynamically stable supramolecular gel it does not fully rule out that the material could be metastable as well. Therefore, we did not dare to make any conclusions on the thermodynamic/kinetic nature of the material. That said, we do agree with the referee that the reader should be aware of the current debate on the thermodynamic/kinetic aspects of supramolecular gelation and have therefore included the following paragraph in the manuscript.
“… Moreover, recent studies indicate supramolecular gelation to be a process controlled by kinetics or thermodynamics depending on the system. Indeed, while the supramolecular gel state was generally believed to be a kinetically-trapped metastable state because of the spontaneous gel-to-crystal transitions observed for multiple systems, a recent theoretical investigation led by Tuttle and Ulijn revealed that supramolecular gel fibers could be at thermodynamic equilibrium as well [16,17]. Clearly, further fundamental studies are required to reach a general overview of the kinetic and thermodynamic effects that drive supramolecular gelation.”
Heat trigger is also not defined. Figure 1 describes a system that gels much faster at elevated temperature than at room temperature. Is this a surprise? Can this be explained using normal kinetic and thermodynamic terms?Why do the authors claim there is no heat at 22C?!
This comment by the referee is related to the definition of triggerless and heat-triggered gelation. Of course, we do not want to claim that there is no heat at 22°C. The difference between heat-triggering and triggerless is related to the change in temperature during the gelation process. For heat-triggering, a change in temperature is induced by first heating the sample (until 100°C) at which the gelator molecules solubilize, followed by letting the sample cool to room temperature (22°C) at which the sol-to-gel transition occurs. Note that the gelling of the system does not take place at a constant (elevated) temperature. In contrast, for the triggerless gelation, the process does take place at constant temperature. We hope that with the added paragraphs (see comments above) this definition is now made more clearer in the manuscript.
It is my impression that the work is worth publishing but the manuscript is still half baked.
Reviewer 4 Report
Urea-based substance that, when mixed with no additional catalyst, spontaneously forms a stable hydrogel.
The triggerless gelation occurs similarly to traditional heat-triggered gelation, according to single particle tracking studies and scanning electron microscopy observations.
The results will be useful for low molecular weight gelators that undergo supramolecular gelation.
I'd suggest writers to substitute more current research with out-of-date ones (1983 and 1969), including https://doi.org/10.1007/s42765-020-00028-w, https://doi.org/10.1016/j.mtadv.2019.100021, and https://doi.org/10.1039/C8MH01130C.
This study can be published in Gels once any small inaccuracies have been thoroughly checked.
Author Response
Urea-based substance that, when mixed with no additional catalyst, spontaneously forms a stable hydrogel.
The triggerless gelation occurs similarly to traditional heat-triggered gelation, according to single particle tracking studies and scanning electron microscopy observations.
The results will be useful for low molecular weight gelators that undergo supramolecular gelation.
I'd suggest writers to substitute more current research with out-of-date ones (1983 and 1969), including https://doi.org/10.1007/s42765-020-00028-w, https://doi.org/10.1016/j.mtadv.2019.100021, and https://doi.org/10.1039/C8MH01130C.
This study can be published in Gels once any small inaccuracies have been thoroughly checked.
We highly appreciate the acknowledgement of the importance and interest in our work for the the field of supramolecular gels by the reviewer. Concerning the mentioned (older) references: J. Phys. Chem. 1969, 73, 34-39, doi:10.1021/j100721a006 reports experimental data on the viscosity of water at different temperatures. This study is crucial to state the sentence in the manuscript: “Note, that the medium’s viscosity increases by several hundred cP during the heat-triggered gelation, meaning the inherent temperature dependence of the viscosity of pure water cannot be the main reason for the observed trends [40]”. For J. Chem. Phys. 1983, 79, 926-935, doi:10.1063/1.445869, this article indicates the value of the TIP3P water model, used during our MD simulations. Hence, both references cannot be substituted. That said, we do feel that the comment of the referee revealed a gap in the introduction of our manuscript. Indeed, there are no recent references included in the manuscript that focus on the applications of supramolecular gels. For this reason, we included among others, the references proposed by the referee in the relevant location of the manuscript.
Round 2
Reviewer 1 Report
The revised paper is well-organized and can be considered for publication though I am still unsatisfied with the novelty of these so-called triggerless supramolecular gels.